# Anticancer and Antioxidant Activities in *Ganoderma lucidum* Wild Mushrooms in Poland, as Well as Their Phenolic and Triterpenoid Compounds

**DOI:** 10.3390/ijms23169359

**Published:** 2022-08-19

**Authors:** Joanna Kolniak-Ostek, Jan Oszmiański, Anna Szyjka, Helena Moreira, Ewa Barg

**Affiliations:** 1Department of Fruit, Vegetable and Plant Nutraceutical Technology, Wrocław University of Environmental and Life Sciences, 37 Chełmońskiego Street, 51-630 Wroclaw, Poland; 2Department of Basic Medical Sciences, Faculty of Pharmacy, Wroclaw Medical University, 211 Borowska Street, 50-556 Wrocław, Poland

**Keywords:** lingzhi, reishi, mushrooms, antioxidants, bioactive compounds, LC-MS

## Abstract

The goal of this study was to the assess anti-cancer and antioxidant properties of the *Ganoderma lucidum* fruiting body, and to identify bioactive compounds found in their extracts. Significant antiproliferative activity was observed against MCF-7, MCF-7/DX, LOVO, LOVO/DX, MDA-MB 231, SW 620, and NHDF cell lines. With IC_50_ values of 25.38 µg/mL and 47.90 µg/mL, respectively, the extract was most effective against MDA-MB 231 and SW 620 cell lines. The bioactive compounds were identified using an ACQUITY UPLC-PDA-MS system. The extracts contained 13 triterpenoids and 28 polyphenols from the flavonols, phenolic acids, flavones, flavan-3-ols, and stilbenes families. Ganoderic acid derivative was found to be the most abundant triterpenoid (162.4 mg/g DW), followed by ganoderic acid B (145.6 mg/g DW). Resveratrol was the most abundant phenolic in the extract (5155.7 mg/100 g DM). The findings could explain why *G. lucidum* extracts are used in folk medicine.

## 1. Introduction

Food’s impact on the human body has received more attention in recent years. Lifestyle diseases are the leading cause of death in both Poland and other European countries. Cancer accounts for almost 27% of all deaths in the European Union [1]. Surgery is the most frequently used method of treating early-stage cancers [2], while later, chemotherapy and radiotherapy, hormone therapy, and immunotherapy are additionally used [3,4]. However, these methods have many side effects, such as high toxicity, pain, emotional stress, negative impact on fertility, and neurological sequelae [5]. In order to alleviate the negative effects and support the effectiveness of the therapies used, biologically active compounds found in medicinal plants can be used. *Ganoderma lucidum* (also known as “the fungus of immortality”) is a traditional medicinal mushroom and one of nature’s most effective adaptogens, acting as a regulator of biological activities. For over a thousand years, the Chinese have used *G. lucidum* for medicinal purposes. *G. lucidum* is still used as an adjuvant treatment for a variety of ailments in China as a traditional herbal medicine. Traditional herbal medicine is thought to have precise clinical effects by reducing inflammation and regulating immunity, as well as antiaging properties [6]. *G. lucidum’s* pharmacological properties stem from its distinct chemical composition, which includes bioactive compounds such as terpenoids, polysaccharides, phenolics, steroids, proteins, glycopeptides, and fatty acids [7,8]. These substances have been linked to *G. lucidum’s* antioxidant, antibacterial, anti-inflammatory, and anti-tyrosinase actions [9,10]. *G. lucidum* has also been used as an adjuvant in the treatment of a number of medical conditions, including insomnia, anorexia, chronic hepatitis, and hypertension [11]. Clinical studies show that the use of *G. lucidum* extracts in combination with routine therapies, such as chemotherapy, radiotherapy, and surgery, supports the patient’s immune system and alleviates the negative effects of chemotherapy and radiotherapy [12,13,14]. The mechanism of action of *G. lucidum* on cancer consists in reducing the vitality of cancer cells, inducing apoptosis, reducing the volume of tumors, and regulating key signaling molecules [15,16,17,18,19].

The purpose of this research was to identify antitumor activity of *G. lucidum*. For this purpose, seven human cancer cell lines (MCF-7, MCF-7/DX, LOVO, LOVO/DX, MDA-MB 231, SW 620, and NHDF) were examined. Next, the methanolic extract of *G. lucidum* fruiting bodies was thoroughly described using UHPLC-MS analysis in terms of specific polyphenolics and triterpenoids. The antioxidant activity of *G. lucidum* was also determined. This is the first time that such a broad characterization of polyphenolic compounds has been performed, with compounds belonging to polyphenolic acids, flavonols, flavanols, flavones, and stilbenes. Our results demonstrate the potential anticancer activity of *Ganoderma lucidum* against breast and colorectal cancer.

## 2. Results and Discussion

### 2.1. Characterization of Phenolic Compounds

Table 1 and Figure 1 show the 28 compounds identified in the *G. lucidum* fruiting body methanolic extract.

The molecules identified in negative ion mode are from the compound group of flavonols, phenolic acids, flavan-3-ols, flavones, stilbenes, and secoiridoids. This is the first time that such a detailed characterization of *G. lucidum* bioactive compounds that are not triterpene compounds has been performed using UPLC–qTOF-MS/MS. Nine phenolic acids and their derivatives (peaks **1**–**4**, **8**, **10**, **20**, **22**, and **25**), five flavonols (peaks **6**, **17**, **18**, **21**, and **23**), six flavan-3-ols (**7** and **12**–**16**), four flavones (**9**, **19**, **27**, and **28**), and one stilbene (peak **24**) were discovered for the first time in *G. lucidum*. The HPLC technique had previously identified chlorogenic acid (peak **5**), quercetin galactoside (peak **11**), quercetin (peak **26**), kaempferol, protocatechuic acid, and caffeic acid in *G. lucidum* extracts [7,9,10,20,21].

#### 2.1.1. Phenolic Acids and Derivatives

Peak **1** has an [M–H]¯ molecular ion at *m*/*z* 827.0046 and was identified as tricaffeoyl-glucosyl-glucoside (Table 1) based on its product ions at *m*/*z* 665, 383, 341, 221, and 179, which is consistent with the data reported by Benayad et al. [22]. Peak **2** was identified as tricaffeoyl-glucosyl with an *m*/*z* of 665.0901. The fragment signals were found in the MS data at *m*/*z* 383, 341, 221, and 128 [22]. Peak **3** was identified as caffeoyl trihexoside because of its [M–H]¯ ion at *m*/*z* 665.0853 and product ion at *m*/*z* 341 (caffeoyl hexose loss of two hexoses). Based on its UV–vis spectra and comparison with the corresponding authentic standard of protocatechuic acid, compound **4** with an *m*/*z* at 315.1666 was tentatively identified as protocatechuic acid hexoside. The MS data revealed a fragment signal at *m*/*z* 153 (hexose loss), which is typical of this compound. The observed fragmentation pathways, as well as the UV spectra and retention times of peaks **5** and **10**, are consistent with authentic standards of 1- and 5-caffeoylquinic acids (Table 1). Peak **8** was identified as trans-5-*p*-coumaroylquinic acid based on its [M–H]¯ ion at *m*/*z* 337.1764 and fragmentation pattern in negative ion mode at *m*/*z* 191 and *m*/*z* 173. (Table 1) [23]. Peak **20** was identified as caffeoyl-2-hydroxyethane-1,1,2-tricarboxylic acid [24]. At *m*/*z* 597.1167, compound **22** exhibits an [M–H]¯ molecular ion, yielding fragments at 359 (rosmarinic acid), 295, and 179. Barros et al. [25] discovered a similar property in yunnaneic acid F. Peak **25** was identified as salvianolic acid B, because of its [M–H]¯ at *m*/*z* 716.9802, which fragmented at *m*/*z* 519, 321, and 295. Barros et al. [25] discovered the same fragmentation pattern for this compound in lemon balm.

#### 2.1.2. Flavonols

The MS/MS fragmentation revealed that the pseudomolecular cations of identified flavonols Peaks **6** and **18** were identified as isorhamnetin 3-*O*-galactoside and isorhamnetin 3-*O*-rutinoside, respectively [23].

Peaks **11**, **17**, and **26** were the precursors of quercetin ([M–H]¯ at *m*/*z* 301). Compound **11** was identified as quercetin 3-*O*-hexoside (Table 1) based on authentic standards of quercetin 3-*O*-galactoside and glucoside, as well as literature data [23]. Peak **17** contained a pseudomolecular ion at *m*/*z* 389.1766 that fragmented at *m*/*z* 301 due to an 88 Da loss. This compound was tentatively identified as a quercetin derivative. Peak **26** contained an [M–H]¯ at *m*/*z* 301.1369. Based on the comparison of its UV–vis spectra with those of corresponding authentic standards, this compound was tentatively identified as quercetin.

Peaks **21** and **23** were kaempferol precursors. Based on their [M–H]¯ at *m*/*z* 796.8967 and 796.8611, respectively, and the characteristic [M–H]¯ at m/z 285, which correspond to kaempferol, they were identified as kaempferol derivatives (Table 1). Furthermore, the UV spectra match those of authentic kaempferol 3-*O*-rutinoside and kaempferol 3-*O*-galactoside standards.

#### 2.1.3. Flavan-3-ols

Because of their [M–H]¯ ions at 577.0848 and 557.0640, compounds **7** and **12** were identified as B-type procyanidin dimers (Table 1), while peaks **14** and **15** were identified as B-type procyanidin trimer and tetramer, respectively, based on their [M–H]¯ at *m*/*z* 864.9185 and 1153.7496. Peak **13** ([M–H]¯ 289.1787 ion) was identified as (-)-epicatechin based on the reference compound. Peak **16** was identified as an A-type procyanidin dimer based on its [M–H]¯ ion at *m*/*z* 575.0462 and the literature [23].

#### 2.1.4. Flavones

Based on the fragmentation pattern found in the literature, compounds **9** and **19** were tentatively identified as diosmetin pentosidine and chrysin-6-*C*-arabinoside-8-*C*-glucoside, respectively [26].

Peaks **27** and **28** were the precursors of apigenin ([M–H]¯ at *m*/*z* 269). Compound **27** was identified as an apigenin derivative (Table 1) based on its *m*/*z* of 313.1686,the characteristic *m*/*z* 269 fragment, and comparison to the authentic apigenin 7-*O*-glucoside standard. Peak **28** contained an [M–H]¯ at *m*/*z* 269.1629, and, based on the comparison to the authentic standard of apigenin 7-*O*-glucoside, this compound was tentatively identified as apigenin.

#### 2.1.5. Stilbenes

Compound **24** with [M–H]¯ at *m*/*z* 227.2041 was identified as resveratrol (Table 1), based on previously published data [27].

### 2.2. Quantitative Analysis of Polyphenols

The *G. lucidum* fruiting bodies had a total phenolic content of 13,991.10 mg/100 g dry weight (DW) (Table 1). Stilbenes (5155.70 mg/100 g DW) were the most abundant group of compounds found in the samples, followed by flavones (total of 4572.51 mg/100 g DW). Resveratrol (compound **24**) and apigenin (compound **28**) were the most abundant compounds in the extract by content (Figure 2).

The phenolic acid content was the lowest, at 912.38 mg/100 g DW (Table 1). Apigenin was the dominant compound in the flavone group, accounting for approximately 88 percent of the group’s content. The A-type procyanidin dimer accounted for approximately 41% of the flavan-3-ols, while the dominant compounds in the flavonols group were quercetin derivatives (about 59 percent). Caffeic acid derivatives predominated in the phenolic acid group, with 1-caffeoylquinic acid having the highest content (~55%).

Various concentrations of polyphenolics have previously been reported in the *G. lucidum* fruiting body. Due to differences in the determination method (application of the Folin–Ciocalteu method) or extraction technique, comparing those data with our results is difficult. Lin et al. [28] used CO_2_ extraction to obtain polyphenolic compounds similar to ours (4131–6376 mg/100 g DW), and Sheikh et al. [21] used the Folin–Ciocalteu method (4601–7143 mg/100 g). Kim et al. [20] obtained much lower levels of polyphenols (16.2 mg/100 g DW). Using the HPLC technique, they identified only ten compounds from the flavonol, phenolic acid, and flavone groups. Furthermore, Dong et al. [7] obtained much lower results (up to 468 mg/100 g DW) in mushrooms dried using various techniques. Yahia et al. [9] discovered nine phenolic compounds amounting to 48.47 mg/100 g DW using HPLC-ESI-MS.

So far, the health-promoting properties of *Ganoderma* fungi have been attributed primarily to the presence of triterpenoid compounds [29]. However, the high polyphenol content found in our study could indicate that this class of compounds plays an important role in shaping the therapeutic properties. Polyphenols, which have antioxidant and anti-inflammatory properties, have a wide range of health-promoting properties [30].

### 2.3. Characterization and Quantification of Triterpenoids

Table 2 and Figure 3 show the 13 triterpenoids identified in the *G. lucidum* fruiting bodies.

Analysis revealed the presence of typical *Ganoderma* species triterpenoid compounds. All triterpenoid compounds were identified using data from the literature.

Peak **1** was identified as ganoderic acid C2 with [M–H]¯ at *m*/*z* 517.3228 and a fragmentation ion at *m*/*z* 499 (Table 2) [29]. Peak **2** contained a pseudomolecular ion at *m*/*z* 529.2790 that fragmented at *m*/*z* 511 and was identified as ganoderic acid C6 [10]. Based on its [M–H]¯ at *m*/*z* 459.2761, fragmentation ion at *m*/*z* 441, and literature data, compound **3** was identified as lucidenic acid N [29]. Peak **4** was identified as ganoderic acid G, with [M–H]¯ at *m*/*z* 531.2991 and fragmentation ions at *m*/*z* 513 and 469 [10]. Based on literature data [8], peak **5** was identified as ganoderenic acid B, with its [M–H]¯ at *m*/*z* 513.2840 and a fragmentation ion at *m*/*z* 495. Compound **6** was identified as ganoderic acid B, with [M–H]¯ at *m*/*z* 515.2890 and fragmentation ion at *m*/*z* 497 [10]. Peak **7** contained a pseudomolecular ion with a mass of 529.2063, which fragmented at *m*/*z* 511 and was identified as a ganoderic acid derivative [10]. According to its fragmentation pathway and literature data, compound **8** was identified as lucidenic acid A [29]. Compound **9**, identified as ganoderenic acid K, had a [M–H]¯ of 571.2933 and a fragmentation ion of 553 [29]. Compound **10** was identified as ganoderic acid AM1 (Table 1) [29]. Peaks **11** and **12**, with [M–H]¯ at *m*/*z* 573.3075 and 569.2740, respectively, were identified as ganoderic acid K and F [8,29]. Compound **13** was identified as ganoderic acid A based on a ganoderic acid authentic standard and the literature [10,31].

The total triterpene concentration determined in the *G. lucidum* fruiting bodies was 769.1 mg/g of extracts DW. Ganoderic acid derivatives (total 478.9 mg/g DW) were the most abundant, followed by ganoderenic acid derivatives (total 308.0 mg/g DW) (Figure 4). The lucidenic acid derivative concentration was 36.8 mg/g DW (Table 1).

These findings are similar to those reported by Lin et al. [28]. Total triterpenoid concentrations in their research on supercritical fluid extraction ranged from 196.03 to 643.06 mg/g DW. The concentration of triterpenoids in the study by Taofiq et al. [10] ranged between 280.46 and 531.21 mg/g, depending on the extraction technique. Bidegain et al. [32] obtained lower results. Triterpenoid compound concentrations in their study ranged between 37.4 and 47.6 mg/g, depending on the cultivation method.

Triterpenoids produced by *G. lucidum* have a variety of biological properties, including antioxidant, antimicrobial, antitumor, anti-hepatitis B, and anti-HIV-1 activity [33]. Their high concentration in *G. lucidum* influences its pharmacological properties, including antiaging, anti-inflammatory, immunomodulatory, and antitumor properties [34,35].

### 2.4. Antioxidant Activity

Because of its high triterpenoid and polyphenolic concentrations, *G. lucidum* has significant antioxidant capacity according to the literature. The antioxidant activity of *G. lucidum* was assessed in this study using the DPPH and ABTS radical scavenging activity as well as the FRAP assay. DPPH, ABTS, and FRAP values were 51.30, 81.26, and 49.87 µmol of Trolox per 1 g DW, respectively, as shown in Table 3. Resveratrol and apigenin, as major components of the extract, as well as ascorbic acid were taken as positive controls.

Ascorbic acid was characterized by the highest value of antioxidant capacity in all the methods used. In the case of the DPPH method, the value of the antioxidant capacity of pure ascorbic acid was over 12 times higher than that of *G. lucidum*. For the ABTS and FRAP methods, the values were about 6 and 10 times higher. The results obtained for apigenin were comparable with those of *G. lucidum* extract. In the case of resveratrol, a higher value of antioxidant capacity was observed compared to the *G. lucidum* methanolic extract. The value was 1.9 times higher for the DPPH method, 11.7 times higher for ABTS, and over 32 times higher for FRAP.

DPPH and ABTS levels were comparable to those previously reported in strawberries [36], while FRAP levels were comparable to those previously reported in oregano, marjoram, and lemon balm (472.32, 463.96, and 464.83 µmol Trolox/1 g DW, respectively) [37].

The content of bioactive compounds, such as polyphenols and triterpenoids, influences antioxidant capacity. Dong et al. [7] discovered a strong correlation between DPPH, ABTS, and FRAP antioxidant capacities and polyphenolic and triterpenoid content in *G. lucidum*. Saltarelli et al. [38] also reported similar findings. The impact of bioactive compounds on antioxidant capacity formation has been discussed in the literature. Procyanidin polymers have a high antioxidant capacity, according to Rice-Evans et al. [39], due to the effects of hydroxylation and glycosylation on the B ring. In a study of the tocopherols found in sour jujube, Qiao et al. [40] discovered that some triterpenoids have more than 15 times the antioxidant activity of ascorbic acid, while others have a low antioxidant activity.

### 2.5. Antiproliferative Activity

The antiproliferative activity of *G. lucidum* was evaluated in ethyl acetate/hexane (1:1, v:v) mixture) after 72 h of incubation with MCF-7, MCF-7/DX, MDA-MB 231, LOVO, LOVO/DX, SW 620, and NHDF cell lines by the MTT assay [41]. The results obtained for breast and colon cancer cells and for normal cells are given in Figure 5, Figure 6 and Figure 7, respectively. The IC_50_ values are summarized in Table 4.

Here we observed that the extract obtained from *G. lucidum* exerted a significant anti-proliferative effect on all tested cancer cell lines in a dose-dependent manner. Importantly, the extract did not exhibit cytotoxicity on normal cells up to 160 μg/mL, which indicates a specific antitumor activity.

The extract was most effective against MDA-MB 231 and SW 620 cell lines with the IC_50_ values of 25.38 μg/mL and 47.90 μg/mL, respectively. In contrast, doxorubicin-resistant MCF-7/DX and LOVO/DX cell lines were found to be less sensitive to the extract at concentrations up to 160 μg/mL. In addition, the results showed that the extract was more effective against drug sensitive MCF-7 and LOVO cells compared to doxorubicin-resistant MCF-7/DX and LOVO/DX cells. The highest extract concentration, of 320 μg/mL, induced significant antiproliferative effects on all tested cancer cells line, achieving growth inhibition over 50% (from 53% up to 91%, depending on cancer cells). However, it should be noted that at this concentration, the extract also caused an important antiproliferative effect on normal cells (inhibition by 40%).

*G. lucidum* is a commonly used herbal medicine in many oriental countries [42] and has been studied by several research groups. Stojković et al. [43] evaluated the cytotoxic effect of two *G. lucidum* extracts from Serbia and China. They reported no cytotoxic effect of the Chinese extract on the HCT15 colon cancer cell line or MCF-7 cell line. It had a cytotoxic effect against MCF-7 cells with GI50 309.66 μg/mL but had no cytotoxic effect on HCT15 cells. In another study, an ethanolic extract containing phenolic compounds showed antiproliferative activity on HeLa (cervical carcinoma cell line), A549 (lung cancer cell line), EA.hy 926 (permanent human cell line derived by fusing human umbilical vein endothelial cells (HUVEC) with human lung adenocarcinoma epithelial cells (A549)), and colon LS174 (colon cancer cell line), as presented by Veljović et al. [44]. The authors found significant correlations between the antiproliferative effect and the total phenolic compounds/glucan content. The most abundant phenols in this extract were hesperetin and naringenin. In our study, a significant part of the antiproliferative activity of our extract towards cancer cells might be attributed to the high content of phenolic compounds: resveratrol had 5155.70 mg and apigenin had 4039.08 mg on 13,991.10 mg of total polyphenolic compounds per 100 g of the extract. Both phytochemicals are known for their potent anticancer activity, as well as chemopreventive and chemoprotective effects [45]. Our data confirm that resveratrol at low concentrations (up to 34 µg/mL) has significant antiproliferative activity on breast and colon cancer cell lines (data not shown).

In addition to phenolic compounds, other phytochemicals may also be responsible for cytotoxicity of the *G. lucidum* extracts towards cancer cells. Raj et al. [46] reported antiproliferative effects of polysaccharides and triterpenoids extracted from *G. lucidum*. In particular, triterpenes known as ganoderic acids are the important fractions responsible for the therapeutic efficacy of *G. lucidum*. Ganoderic acids are capable of inducing apoptosis and autophagy in various cancer cell types [47]. Our extract contains 13 different triterpenes in the amount of 769.1 mg per g of the extract, which may act synergically with phenolic compounds in anticancer effects. Naturally occurring compounds, in addition to their direct effect on cancer cells, often exhibit immunomodulatory effects, which may also induce an anti-neoplastic response [48]. *Ganoderma* has been shown to be a promising anticancer immunotherapy agent. Together with relatively low toxicity, the extract may be used in combination therapy or as a dietary supplement [49].

## 3. Materials and Methods

### 3.1. Reagents and Standards

Acetonitrile, methanol, formic acid, DPPH (1,1-diphenyl-2 picrylhydrazyl radical), ABTS (2,2-azinobis(3-ethylbenzothiazoline-6-sulphonic acid) radical cation), TPTZ (2,4,6-tri(2-pyridyl)-s-triazine), Trolox (6-hydroxy-2,5,7,8-tetramethylchroman-2-carboxylic acid), FeCL_3_, acetic acid, ganoderic acid A, caffeic acid, and resveratrol were purchased from Sigma-Aldrich (Steinheim, Germany). Apigenin 7-*O*-glucoside, isorhamnetin 3-*O*-glucoside, kaempferol 3-*O*-glucoside, quercetin 3-*O*-glucoside, chlorogenic, *p*-coumaric, quinic, ferulic and rosmarinic acids, procyanidin B2, and (-)-epicatechin were purchased from Extrasynthese (Lyon, France). All reagents were of analytical grade. DMEM/F12 (Dulbecco’s Modified Eagle’s Medium/Nutrient Mixture F-12), DMEM (Dulbecco’s Modified Eagle’s Medium), HBSS (Hank’s Balanced Salt Solution), FBS (fetal bovine serum), UltraGlutamine 1, and gentamicin sulfate were purchased from Lonza (Basel, Switzerland). TrypLE Express and MEM (Minimum Essential Media) were from Gibco (Waltham, MA, USA). MTT ((3-(4,5-dimethylthiazol-2-yl)-2,5-diphenyltetrazolium bromide) and DMSO (dimethyl sulfoxide) were purchased from Sigma-Aldrich (St. Louis, MO, USA). Isopropanol was purchased from Chempur (Piekary Śląskie, Poland).

### 3.2. Plant Materiaµl

In July 2020, fresh *G. lucidum* fruiting bodies were hand-harvested from a private forest in Szczytna (50°24′47″N 16°26′50″E), Lower Silesia, Poland. Fresh mushrooms were cut, frozen in liquid nitrogen, freeze-dried (24 h; Christ Alpha 1–4 LSC, Martin Christ GmbH, Osterode am Harz, Germany), and crushed in a closed laboratory mill to produce homogeneous powders (IKA 11A, Staufen, Germany).

### 3.3. Extraction Procedure

Phenolic compounds were isolated from the lyophilized powder by extraction with 80% ethanol. The ratio of this solvent to the raw material was 3:1 (*v*/*v*). After filtration, the solution was concentrated with a vacuum pump and freeze-dried and then analyzed. A total of 0.1 g of sample was mixed with 5 mL of 30% UPLC-grade methanol, sonicated (20 min), centrifuged (19,000 g/10 min), and filtered (hydrophilic PTFE 0.20 µm membrane (Millex Samplicity Filter, Darmstadt, Germany)).

### 3.4. UPLC-DAD-ESI/MS Analysis of Polyphenols

The analysis was carried out according to Kolniak-Ostek [50], on an ACQUITY UPLC-DAD-ESI/MS instrument (Waters Corp., Milford, MA, USA). A UPLC BEH C18 column (1.7 µm, 2.1 mm × 100 mm; Waters Corp., Milford, MA, USA) was used. The mobile phase was made up of aqueous 0.1% formic acid (A) and 100% acetonitrile (B).

### 3.5. UPLC-DAD-ESI/MS Analysis of Triterpenoids

Extraction, identification, and quantification of triterpenoids were performed as described previously by Kolniak-Ostek [36]. A total of 10 mL of ethyl acetate/hexane (1:1, *v*:*v*) mixture was used to extract 0.5 g of sample. The analysis was carried out on ACQUITY UPLC-DAD-ESI/MS instrument. The separation was carried out on a Waters Corp. UPLC BEH C18 column (1.7 µm, 2.1 mm × 150 mm). The mobile phase was made up of 100% methanol (A) and 100% acetonitrile (B) (15:85, *v*/*v*).

### 3.6. Determination of Antioxidant Activity

The DPPH, ABTS, and FRAP assays were performed in the same manner as previously described by Yen and Chen [51], Re et al. [52], and Benzie and Strain [53], respectively. All measurements were taken on a Synergy H1 microplate reader (BioTek, Winooski, VT, USA). The standard curve was created by varying the concentrations of Trolox. As positive controls, resveratrol, apigenin, and ascorbic acid in concentrations of 1mg/mL were determined. The results are expressed in terms of µMol of Trolox equivalents (TE) per 1 g of extract (µmol TE/1 g).

### 3.7. Determination of Antiproliferative Activity In Vitro

#### 3.7.1. Cell Lines and Culture Conditions

The study used seven cell lines: MCF-7, MCF-7/DX (doxorubicin-resistant subline of MCF-7 cells), and MDA-MB-231 from breast cancer; LOVO, LOVO/DX (doxorubicin-resistant subline of LOVO/DX), and SW620 from colorectal cancer; and normal human dermal fibroblast from colorectal cancer (NHDF). The ATCC collection was used to obtain all cell lines. LOVO/DX and MCF-7/DX were created by cultivating parental counterparts in the presence of low concentrations of doxorubicin (DX). Cultures were grown in a humidified incubator with 5% CO_2_, 37 °C, and 95% humidity. The cell lines MCF-7/DX, MCF-7, MDA-MB-231, and NHDF were cultured in DMEM medium supplemented with 10% FBS, 2 mM L-glutamine, and 20 g/mL gentamicin sulfate. LOVO, LOVO/DX, and SW620 cell lines were grown in DMEM/F12 medium with the aforementioned supplements. At 37 °C, cells were passaged twice a week with TrypLE Express solution, then reduced and resuspended in culture medium.

#### 3.7.2. Tested Compound

*G. lucidum* extract was dissolved in DMSO to make a stock solution of 64 mg/mL and stored at −20 °C. Before each experiment, a 100× dilution of the stock solution in a culture medium was used to make the working solution. The cells were subjected to serial dilutions of the tested extract (at final concentrations of 20, 40, 80, 160, and 320 g/mL). In the highest concentration of the extract, the final DMSO concentration in the cell culture did not exceed 0.5 percent.

#### 3.7.3. MTT Assay

The MTT tests were carried out in accordance with the standard procedure outlined in the literature [54]. Using TrypLE solution, the cells were detached from the culture bottle surface (when confluency was greater than 70%), counted, and resuspended in full culture medium. The cells were then seeded in 96-well plates at 2.5 × 104 cells per well and incubated for 24 h. The cells were treated with various concentrations of the tested extract the next day and incubated for another 72 h. The medium was removed from the wells after incubation, and 50 µL of 1 mg/mL MTT solution in MEM was added to each well. Plates were placed in an incubator (at 37 °C) for 2 h. For 30 min in the dark, formazan crystals were solubilized in 100 µL of isopropanol. Finally, absorbance at 555 nm was determined using a Wallac 1420 Victor2 microplate reader (PerkinElmer, Waltham, MA, USA).

The extract’s antiproliferative effect was compared to control cell cultures (without the extract) and expressed as E/E0 × 100 percent values. The IC_50_ values were calculated based on the dose-response curves. The IC_50_ is defined as the concentration of a tested extract that inhibits cell growth by 50%.

#### 3.7.4. Statistical Analyses

The data of antiproliferative activity were expressed as the mean ± standard deviation of eight replicates from three independent experiments. The *t*-test was used to perform statistical analysis on the experimental data. GraphPad Prism 6 software was used for statistical analysis (InStat Software, San Diego, CA, USA). At the confidence level of *p* ˂ 0.05, significant differences between each set of data were considered. For bioactive compounds and antioxidant activity, the data were expressed as the mean ± standard deviation of three replicates. Statistical analysis was conducted using Statistica 13.1 software (StatSoft, Kraków, Poland). Significant differences (*p* ≤ 0.05) between means were evaluated by one-way ANOVA and Duncan’s test.

## 4. Conclusions

In this study, anticancer and antioxidant activities as well as bioactive compounds of the *Ganoderma lucidum* fruiting body were evaluated. The results demonstrate that the extract from *G. lucidum* is a rich source of bioactive components such as phenolic compounds and triterpenoids. Our study provides significant insight into the antiproliferative effects of phenolic compounds and triterpenoids contained in the extract of *Ganoderma lucidum* on breast cancer cell lines (MCF-7, MCF-7/DX, MDA-MB-231), a colorectal cancer cell line (SW 620), and colon cancer cell lines (LOVO, LOVO/DX). The anticancer activity of these compounds should be further investigated in terms of causing cancer cell death through various pathways that could help develop therapies against colon, colorectal, and breast cancer. This shows that *G. lucidum* extracts could be useful in the fight against diseases related to oxidative stress such as cancer.

## Figures and Tables

**Figure 1 ijms-23-09359-f001:**
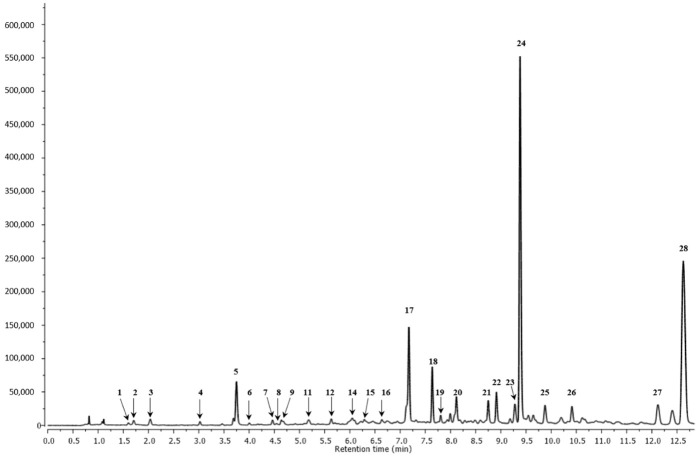
UPLC chromatogram of polyphenolics profile of *G. lucidum* extract at 280 nm. Peak number identities are displayed in Table 1.

**Figure 2 ijms-23-09359-f002:**
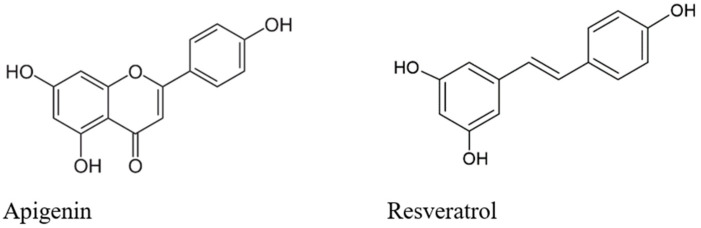
Selected chemical constituents of *G. lucidum* methanolic extract, from the group of phenolic compounds.

**Figure 3 ijms-23-09359-f003:**
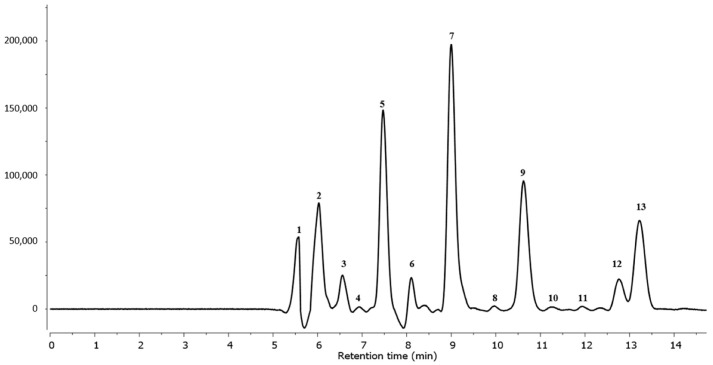
UPLC triterpenoids profile of *G. lucidum* extract at 254 nm. Peak number identities are displayed in Table 2.

**Figure 4 ijms-23-09359-f004:**
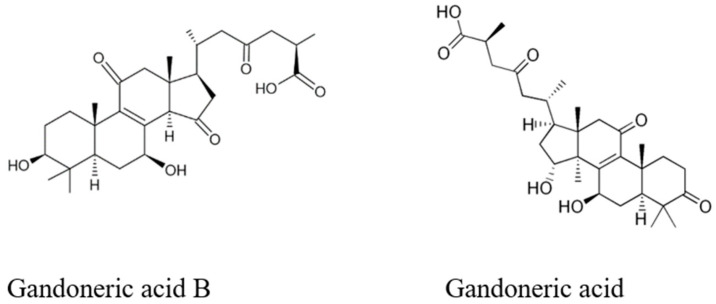
Selected chemical constituents of *G. lucidum* methanolic extract from the group of triterpenoids.

**Figure 5 ijms-23-09359-f005:**
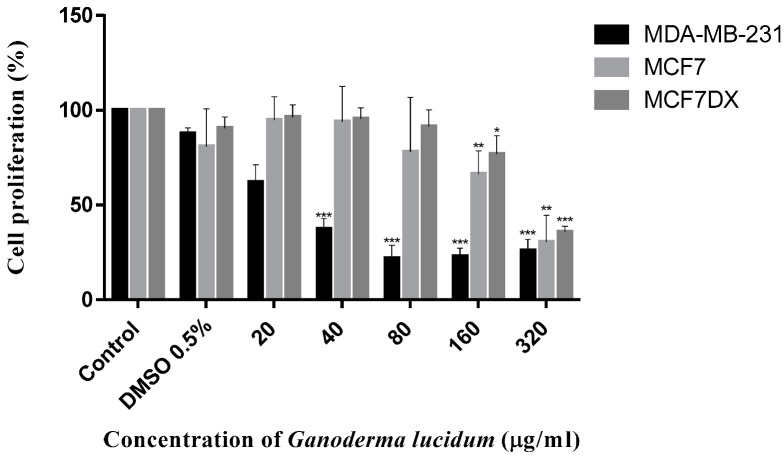
Antiproliferative effect of *Ganoderma lucidum* extract on MDA-MB-231, MCF7, and MCF7/DX after 72 h treatment. The results are the mean ± SD of three independent experiments. The significance of the differences was determined by Student’s *t*-test. * *p* < 0.05, ** *p* < 0.01, *** *p* < 0.001.

**Figure 6 ijms-23-09359-f006:**
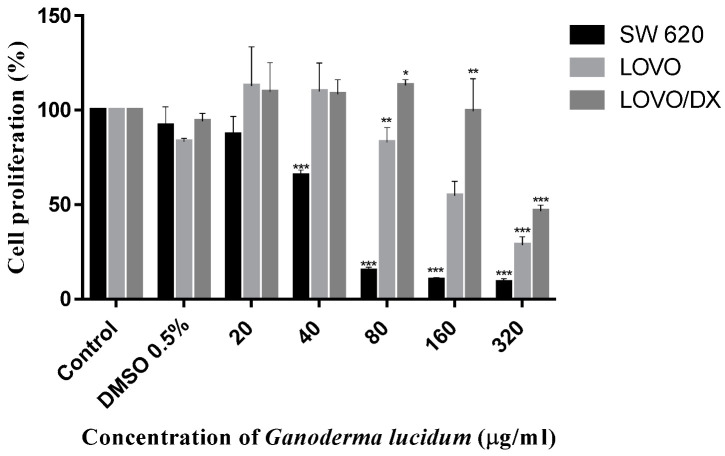
Antiproliferative effect of *Ganoderma lucidum* extract on SW 620, LOVO, and LOVO/DX after 72 h treatment. The results are the mean ± SD of three independent experiments. The significance of the differences was determined by Student’s *t*-test. * *p* < 0.05, ** *p* < 0.01, *** *p* < 0.001.

**Figure 7 ijms-23-09359-f007:**
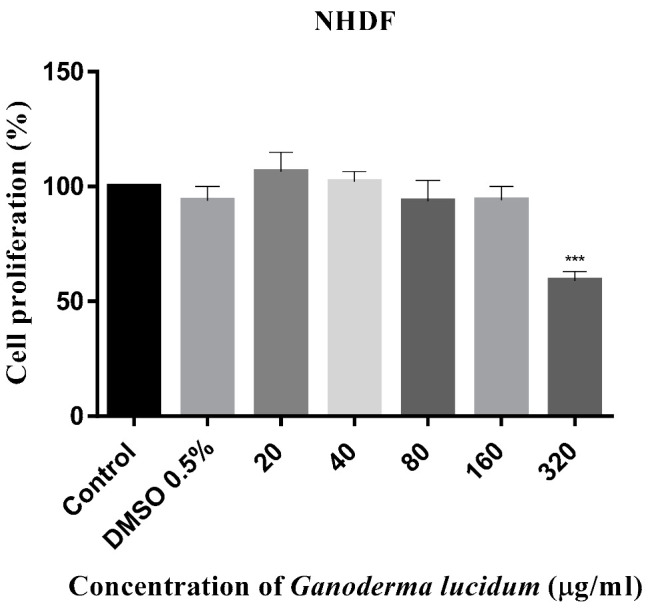
Antiproliferative effect of *Ganoderma lucidum* extract on NHDF cells after 72 h treatment. The results are the mean ± SD of three independent experiments. The significance of the differences was determined by Student’s *t*-test. *** *p* < 0.001.

**Table 1 ijms-23-09359-t001:** Profile and quantity (mg/100g DW) of phenolic compounds in extract of *Ganoderma lucidum*.

No.	Tentative Identification	Rt (min)	UV-Vis (nm)	MS [H−M]^−^ (*m*/*z*)	MS/MS Fragments (*m*/*z*)	Quantity (mg/100 g DW of Extract) ^2^
*Phenolic acids*						
1.	Tricaffeoyl-glucosyl-glucoside	1.60	321	827.0046	665/383/341/221/179	13.54 ± 0.23 f
2.	Tricaffeoyl-glucosyl	1.70	320	665.0901	383/341/221/128	23.79 ± 0.24 e
3.	Caffeoyl trihexoside	2.01	325	665.0853	503/341/179/135	38.02 ± 0.30 d
4.	Protocatechuic acid hexoside	3.02	259	315.1666	153	19.09 ± 0.15 f
5.	1-Caffeoylquinic acid ^1^	3.73	327	353.1640	191/179	505.89 ± 3.21 a
8.	trans-5-*p*-Coumaroylquinic acid	4.55	309	337.1764	191/163	0.46 ± 0.00 h
10.	5-Caffeoylquinic acid ^1^	4.70	324	353.1638	191	95.01 ± 0.92 c
20.	Caffeoyl-2-hydroxyethane-1.1.2-tricarboxylic acid	8.13	326	339.1338	295/251	213.89 ± 1.52 b
22.	Yunnaneic acid F	8.91	275	597.1167	359/295/179	1.29 ± 0.00 g
25.	Salvianolic acid B	9.88	254/287/308	716.9802	519/321/295	1.39 ± 0.00 g
					*Sum*	912.38 ± 20.14 D
*Flavonols*						
6.	Isorhamnetin-galactoside	4.00	350	447.1734	315	18.23 ± 0.65 f
11.	Quercetin hexoside ^1^	5.17	355	463.1020	301	30.41 ± 1.51 e
17.	Quercetin derivative	7.18	354	389.1766	301	818.29 ± 4.17 a
18.	Isorhamnetin 3-*O*-rutinoside	7.64	351	623.1027	315	110.50 ± 1.00 cd
21.	Kaempferol derivative	8.77	346	796.8967	519/285	278.53 ± 1.21 b
23.	Kaempferol derivative	9.28	350	796.8611	519/285	281.38 ± 2.01 b
26.	Quercetin	10.41	340	301.1369	-	133.13 ± 1.99 c
					*Sum*	1670.46 ± 35.15 C
*Flavan-3-ols*						
7.	Procyanidin dimer ^1^	4.46	279	577.0848	425/289	151.10 ± 1.11 cd
12.	Procyanidin dimer ^1^	5.63	280	577.0640	289	175.71 ± 2.08 c
13.	(-)-epicatechin ^1^	5.70	280	289.1787	245	64.20 ± 1.03 e
14.	B-type procyanidin trimer	6.05	281	864.9185	575/285	447.72 ± 2.22 b
15.	B-type procyanidin tetramer	6.29	279	1153.7496	577/407/289	141.48 ± 1.44 d
16.	A-type procyanidin dimer ^1^	6.63	277	575.0462	289	699.84 ± 3.65 a
					*Sum*	1680.05 ± 34.77 C
*Flavones*						
9.	Diosmetin-pentoxide	4.65	342	431.1876	299	83.21 ± 1.57 c
19.	Chrysin-6-*C*-arabinoside-8-*C*-glucoside	7.80	341	547.0973	457/367/337	35.37 ± 1.02 d
27.	Apigenin derivative	12.12	340	313.1686	269	414.85 ± 2.98 b
28.	Apigenin	12.62	340	269.1629	-	4039.08 ± 35.12 a
					*Sum*	4572.51 ± 45.69 B
*Stilbenes*						
24.	Resveratrol	9.38	305	227.2041	185/183/159/157/143	5155.70 ± 41.87 A
					** *TOTAL* **	**13,991.10** ± 98.24

^1^ Identification confirmed by commercial standards. ^2^ Values (mean of three replications) ± standard deviation followed by different letters (a–h) within the same group of compounds are different (*p* = 0.05) according to Duncan’s test. Values (sum of compounds) followed by different letters (A–D) are different (*p* ≤ 0.05) according to Duncan’s test.

**Table 2 ijms-23-09359-t002:** Profile and quantity (mg/g DW) of triterpenoids in extract of *Ganoderma lucidum*.

No.	Rt (min)	UV-vis (nm)	MS [H−M]^−^ (*m*/*z*)	MS/MS Fragments (*m*/*z*)	Tentative Identification	Quantity ^2^ (mg/g DW of Extract)
1.	5.61	266	517.3228	499	Ganoderic acid C2	47.0 ± 0.1 e
2.	5.99	257	529.2790	511	Ganoderic acid C6	71.5 ± 0.2 d
3.	6.55	259	459.2761	441	Lucidenic acid N	24.4 ± 0.1 g
4.	7.47	256	531.2991	513/469	Ganoderic acid G	16.7 ± 0.0 h
5.	8.09	248	513.2840	495	Ganoderenic acid B	145.6 ± 0.6 b
6.	8.69	250	515.2980	497	Ganoderic acid B	28.4 ± 0.0 fg
7.	8.99	256	529.2063	511	Ganoderic acid derivative	162.4 ± 0.6 a
8.	9.95	259	457.2499	441	Lucidenic acid A	12.4 ± 0.1 i
9.	10.62	255	571.2933	553	Ganoderenic acid K	107.7 ± 0.5 c
10.	11.23	261	513.2844	495/451/433/247	Ganoderic acid AM1	10.6 ± 0.0 i
11.	11.98	254	573.3075	555	Ganoderic acid K	11.9 ± 0.0 i
12.	12.79	254	569.2740	551	Ganoderic acid F	31.2 ± 0.1 f
13.	13.25	254	515.3001	497	Gandoneric acid A ^1^	99.2 ± 0.3 c
					TOTAL	769.1 ± 1.2

^1^ Identification confirmed by commercial standards. ^2^ Values (mean of three replications) ± standard deviation followed by different letters (a–i), are different (*p* ≤ 0.05) according to Duncan’s test.

**Table 3 ijms-23-09359-t003:** Antioxidant activity in a methanolic extract of *Ganoderma lucidum* and authentic standards of resveratrol, apigenin, and ascorbic acid.

	DPPH	ABTS	FRAP
	(µMol TE/g)	(µMol TE/g)	(µMol TE/g)
*Ganoderma lucidum*	51.3 ± 1.04	81.26 ± 1.10	49.87 ± 1.58
Resveratrol	444.54 ± 23.07	954.24 ± 3.87	1598.45 ± 5.09
Apigenin	69.76 ± 7.19	91.19 ± 5.10	56.50 ± 1.77
Ascorbic acid	648.45 ± 18.92	480.37 ± 8.60	533.79 ± 12.83

Means ± SD (*p* ≤ 0.05; n = 3).

**Table 4 ijms-23-09359-t004:** Inhibition concentrations (IC_50_; µg/mL) of *Ganoderma lucidum* against six cancer cell lines. Values are presented as mean ± SD.

IC_50_ (µg/mL)	
* **Extract of Ganoderma lucidum** *	
**MDA-MB-231**	25.38 ± 0.24
**MCF7**	209.6 ± 0.24
**MCF7/DX**	235.4 ± 0.26
**SW 620**	47.90 ± 2.60
**LOVO**	188.4 ± 0.76
**LOVO/DX**	*314.9* ± 35.75

## Data Availability

Data are contained within the article.

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
