# Peer review of "Anticancer and Antioxidant Activities in Ganoderma lucidum Wild Mushrooms in Poland, as Well as Their Phenolic and Triterpenoid Compounds"

_ijms, 2022, doi:10.3390/ijms23169359_

Round 1
Reviewer 1 Report
In the title manuscript " Anticancer and antioxidant activities in Ganoderma lucidum wild mushrooms in Poland, as well as their phenolic and triterpenoid compounds" The authors reported the extraction of phenolic as well as triterpenoid compounds from G. lucidum in poland, they tested the extract against cancer and anti-oxidant activities. The study needs more experiments to assist the conclusion of this paper as follow:
1. The author must show the main constituents of the extracts and assign the chemical structures of these species which are crucial to know which compounds are responsible for the reactivity.
2. Further study about cancer activity such as cell cycle assay, apoptosis, and necrosis induction.
3. Possible mechanism of the cancer reactivity must study and confirmed by experimental assays.
4. They should study the ROS generation too for the anti-oxidant activity.
5. The introduction is superficial and must improve.
Author Response
- The author must show the main constituents of the extracts and assign the chemical structures of these species which are crucial to know which compounds are responsible for the reactivity.
Reply: Chemical structures have been added
- Further study about cancer activity such as cell cycle assay, apoptosis, and necrosis induction.
Reply: We are thankful for this remark. We fully agree with the Reviewer that is very interesting and attractive to assess the capacity of the extract to induce apoptosis/necrosis of cancer cells and cell cycle arrest. These studies would provide additional information on cytotoxic / cytostatic effects. However, here we focused on the general study of anticancer potential of tested extract against colon and breast cancers. The results presented in this manuscript are the first evaluation of such effects. Further research will be carried out to evaluate the specific mechanism by which the extract exerts these effects, including studies of types of death and cell cycle arrest. It will be the subject of our another project and paper.
- Possible mechanism of the cancer reactivity must study and confirmed by experimental assays.
Reply: We appreciate this suggestion. As mentioned earlier, this is the goal of our further research.
- They should study the ROS generation too for the anti-oxidant activity.
Reply: Unfortunately, this analysis is not performed in our laboratory. We are also unable to validate the method and perform the indicated tests within 10 days. We will apply your suggestion in future research.
- The introduction is superficial and must improve.
Reply: Introduction has been improved

Reviewer 2 Report
The manuscript describes anti-cancer and antioxidant properties of bioactive compounds found in the extracts of Ganoderma lucidum fruiting body . Significant anti-proliferative activity was observed against a panel of six different cancer cell lines and normal NHDF cell lines. I have these comments on the manuscript that should be addressed before the publication of the manuscript:
a) Only few compounds were confirmed by commercial standards. Most of reported compounds were identified only by UPLC–qTOF-MS/MS based on its product ions and compared with the data reported in literature.
b) Clinical studies on Ganoderma lucidum on miscellaneous tumors (breast, lung, colorectal, and nasopharyngeal cancer) are in progress. Several biologically active compounds of Ganoderma lucidum have been previously identified: Martínez-Montemayor, M.M.; Ling, T.; Suárez-Arroyo, I.J.; Ortiz-Soto, G.; Santiago-Negrón, C.L.; Lacourt-Ventura, M.Y.; Valentín- Acevedo, A.; Lang, W.H.; Rivas, F. Identification of biologically active Ganoderma lucidum compounds and synthesis of improved derivatives that confer anti-cancer activities in vitro. Front. Pharmacol. 2019, 10, 115
c) I’m very surprise that several biological active constituents (ergosterol, 5,6-dehydroergosterol and ergosterol peroxide) that exhibited significant in vitro anti-cancer activities have not been identified in the extracts.
d) A previous article on Ganoderma lucidum fruiting body have been published and not cited. Am J Transl Res. 2020 Jun 15;12(6):2675-2684.
e) In the antioxidant and antiproliferative studies, no positive control was reported. As major component of extracts, resveratrol and apigenin should be employed as positive controls.
f) In the antioxidant and antiproliferative studies, please underline the type of extract employed in this study (by extraction with 80% ethanol or ethyl acetate:hexane (1:1, v:v) mixture).
Author Response
- Only few compounds were confirmed by commercial standards. Most of reported compounds were identified only by UPLC–qTOF-MS/MS based on its product ions and compared with the data reported in literature.
Reply: We only have a few compound standards in our collection that have been identified in Ganoderma Lucidum. The remaining compounds were identified on the basis of the molecular weight, the emerging molecular ions, fragmentation paths and UV spectra found in the literature. Therefore, in Tables 1 and 2 we have indicated that it is 'Tentative identification'
- Clinical studies on Ganoderma lucidum on miscellaneous tumors (breast, lung, colorectal, and nasopharyngeal cancer) are in progress. Several biologically active compounds of Ganoderma lucidum have been previously identified: Martínez-Montemayor, M.M.; Ling, T.; Suárez-Arroyo, I.J.; Ortiz-Soto, G.; Santiago-Negrón, C.L.; Lacourt-Ventura, M.Y.; Valentín- Acevedo, A.; Lang, W.H.; Rivas, F. Identification of biologically active Ganoderma lucidum compounds and synthesis of improved derivatives that confer anti-cancer activities in vitro. Front. Pharmacol. 2019, 10, 115
Reply: Information has been added to manuscript. Citation has been added
- I’m very surprise that several biological active constituents (ergosterol, 5,6-dehydroergosterol and ergosterol peroxide) that exhibited significant in vitro anti-cancer activities have not been identified in the extracts.
Reply: This may be due to the very low concentration of these compounds in the tested extracts or their absence in Ganoderma lucidum growing in Poland. Another reason for the absence of these compounds may be the extraction technique used (80% ethanol + ultrasound).
- A previous article on Ganoderma lucidum fruiting body have been published and not cited. Am J Transl Res. 2020 Jun 15;12(6):2675-2684.
Reply: Citation has been added
- In the antioxidant and antiproliferative studies, no positive control was reported. As major component of extracts, resveratrol and apigenin should be employed as positive controls.
Reply: Results have been added. Regarding the antiproliferative research, we have tested resveratrol in several concentrations in our laboratory and have shown that it has a dose-dependent antiproliferative effect on all cell lines tested. No antiproliferative effect was detected on NHDF cells. However, we are currently obliged not to publish these results due to several collaborative relationships. We have added this information in the text.
Unfortunately, we can’t perform analysis of apigenin or other positive controls within 10 days.
We appreciate this suggestion and will apply these controls in our further research.
- In the antioxidant and antiproliferative studies, please underline the type of extract employed in this study (by extraction with 80% ethanol or ethyl acetate:hexane (1:1, v:v) mixture).
Reply: Information has been added

Round 2
Reviewer 1 Report
The authors have been addressed my comments partially and but the most comments they will consider in the future work which is acceptable.
Reviewer 2 Report
The authors have properly answered to all my questions. I was very surprise abou the response at the point e related to the antiproliferative activity of resveratrol "However, we are currently obliged not to publish these results due to several collaborative relationships." A huge amount of preclinical studies investigated anticancer properties of resveratrol in a plethora of cellular and animal models.